# Comparison of Methyl Bromide and Ethyl Formate for Fumigation of Snail and Fly Pests of Imported Orchids

**DOI:** 10.3390/insects14010066

**Published:** 2023-01-10

**Authors:** Tae-Hyung Kwon, Dong-Bin Kim, Byung-Ho Lee, Dong H. Cha, Min-Goo Park

**Affiliations:** 1Institute of Quality & Safety Evaluation of Agricultural Products, Kyungpook National University, Daegu 41566, Republic of Korea; 2Komohana Research Center, University of Hawaii, Manoa 875, Hilo, HI 96720, USA; 3Pacific Basin Agricultural Research Center, US Department of Agriculture-Agricultural Research Service, Hilo, HI 96720, USA; 4Department of Bioenvironmental Chemistry, Jeonbuk National University, Jeonju 54896, Republic of Korea

**Keywords:** fumigant, imported orchids, methyl bromide alternative, quarantine treatment

## Abstract

**Simple Summary:**

Invasive snails and flies are among the major groups of pests intercepted from imported orchids in Korea, which are controlled by methyl bromide (MB) fumigation. As a first step to develop an alternative treatment, we compared the efficacy and phytotoxicity of ethyl formate (EF) and MB on four species of imported orchids using juvenile stages of *Achatina fulica* and third and fourth instars of *Lycoriella mali*. Efficacy trials showed that EF was at least as effective as MB with LCt_99_ (lethal concentration × time product required for 99% mortality) values of EF at 68.1 and 73.1 g h/m^3^ at 15 °C and LCt_99_ of MB at 95.9 and 78.4 g h/m^3^ at 15 °C for *A. fulica* and *L. mali*, respectively. In scale-up trials, EF treatment at 35 g/m^3^ for 4 h at 15 °C resulted in complete control of *A. fulica* and *L. mali*. MB treatment based on the current treatment guideline for imported orchids (48 g/m^3^, 2 h at >15 °C) resulted in complete control of *L. mali* but not *A. fulica,* which could be completely controlled with 3 h treatment. Leaf chlorophyll contents and hue values of treated orchids were not affected by EF treatment but were significantly changed by MB. All four species of orchid died within 30 d of MB treatment, while only one species could not recover the damage from EF treatment. Our results suggest that EF is as effective as MB for snails and flies and less phytotoxic than MB to imported orchids and may be applicable as an alternative to MB in phytosanitary treatments of invasive snails and flies of imported orchids.

**Abstract:**

Invasive snails and flies are major pests of imported orchids, controlled by methyl bromide (MB) fumigation in Korea. We compared the efficacy and phytotoxicity of ethyl formate (EF) and MB on four species of imported orchids using juvenile stages of *Achatina fulica* and third and fourth instars of *Lycoriella mali*. EF was as effective as MB. The LCt_99_ values of EF were 68.1 and 73.1 g h/m^3^ at 15 °C; and those of MB were 95.9 and 78.4 g h/m^3^ at 15 °C for *A. fulica* and *L. mali*, respectively. In the scale-up trials, EF treatment at 35 g/m^3^ for 4 h at 15 °C resulted in complete control of both pests. MB treatment based on the current treatment guidelines for imported orchids (48 g/m^3^, 2 h, at >15 °C) resulted in complete control of *L. mali* but not of *A. fulica*. Chlorophyll content and hue values of treated orchids were not affected by EF treatment but significantly changed by MB (*p*-value < 0.05). All four treated species of orchids died within 30 d of MB treatment, while only one species died from EF treatment. Our results suggest that EF is a potential alternative to MB in phytosanitary treatment of imported orchids.

## 1. Introduction

Orchids are one of the most highly valued and traded tropical commodities worldwide [1]. More than 1.1 billion bare-rooted and potted orchids and 31 million kg of cut flowers were globally traded between 1995 and 2015; most of these were exported from Taiwan and Thailand to Korea (40%), the United States (27%) and Japan (20%) [1,2]. This trend continued, and, in the year 2021, Korea imported 1.4 million kg of orchids valued at a total of $16 million [3]. Upon arrival at ports in Korea, the orchids are inspected for exotic pests and fumigated with methyl bromide (MB) once the pests are intercepted [4]. Snails (*Achatina* spp., *Oxychilus* spp., and *Bradybaena* spp.) and small flies (*Lycoriella* spp., *Megaselia* spp., and *Bradysia* spp.) are among the main groups of pests intercepted from imported orchids and other nursery products [5,6,7]. These accounted for approximately 44% of orchid pest interception in 2021 [8].

Although the use of MB has been discontinued due to its adverse effects on ozone layer depletion and human health, a critical use exemption by the United Nations Environment Programme still allows MB fumigation for quarantine and pre-shipment (QPS) purposes [9,10,11,12,13]. This is due to a general lack of feasible alternative treatments for pest disinfection in numerous commodities [13,14] and the need for MB to avert trade disruptions. However, an approved MB treatment for a commodity can be phytotoxic [15], and hence the development of alternative treatments to replace MB has become crucial. Thus, this study aimed to conduct a comparative study of MB and an alternative fumigant to evaluate their efficacy to disinfect snails and small fly species and observe their effects on orchid quality, as a first step towards developing an alternative treatment for interception of pests in imported orchids.

Phosphine is a traditionally established effective alternative disinfectant to MB for nursery plants [16,17,18]. However, the effective use of phosphine generally requires a long fumigation time (at least >24 h) [16,19,20,21], which may not be ideal for orchids (especially cut orchid flowers) that require quick treatment to maintain quality and shelf life. Ethyl formate (EF) is another alternative for MB that is considered for the fumigation of imported orchids. It has shown similar efficacy as MB [22,23], requires a short fumigation period [24], breaks down rapidly to ethanol and formic acid without residue [22], and is much safer for the workplace than MB (i.e., exposure limit of EF vs. MB = 100 vs. 1 ppm) [25]. It has been designated as a “generally regarded as safe” (GRAS) chemical by the US Food and Drug Administration, is used in food flavoring [26], and is considered a safer alternative to MB [22,27]. Similar to most known fumigants, EF fumigation can cause phytotoxicity depending on the plant species or cultivars treated and the EF dose applied [16,23,24]. For example, when nursery plants were treated with EF, EF phytotoxicity was observed ranging from none (e.g., *Ficus benghalensis, Pachira macrocarpa*) to severe (e.g., *Anthurium andraeanum, Chamaedorea elegans*) [15]. Recently, 4 h EF fumigation was found to be effective at disinfesting invasive pests on a wide variety of products including food commodities, such as banana, citrus, dry dates, and blueberries [22,23,28,29], and non-food commodities including imported cut flowers and nursery plants [16,30].

In this study, we evaluated the feasibility of using EF as an MB alternative for the fumigation of snails and flies in four different species of bare-rooted imported orchids (*Cymbidium sinense, C. goeringii, Phalaenopsis aphrodite* and *Agave attenuata*). *Achatina fulica* (Stylommatophora: Achatinidae), the giant African land snail (referred to here as “snails”), and *Lycoriella mali* (Diptera: Sciaridae), a mushroom sciarid fly (referred to here as “flies”) were selected as representative invasive snail and small fly species, respectively. MB and EF were tested at LCt_99_ (lethal concentration × time product required for 99% mortality) doses determined for *A. fulica* and *L. mali*. Specifically, we (1) determined the efficacy of EF on the third- and fourth-instar larvae of *L. mali* as they are the most tolerant life stages to EF and juvenile stages of *A. fulica* as they are the most abundant snail stages intercepted from orchids [31] in small-scale laboratory trials (in 6.8 L desiccators) at 15 °C; (2) evaluated EF and MB sorption in four different species of imported orchids to determine an appropriate loading ratio for scale-up trials; (3) conducted commercial-scale EF and MB trials at both 30 and 150 m^3^ based on the results from small-scale laboratory trials and sorption tests; and (4) compared the effects of EF and MB fumigation on the chlorophyll content, hue value, and overall damage of treated orchids.

## 2. Materials and Methods

### 2.1. Snails and Flies

*L. mali* was captured from a commercial mushroom farm in Yeongcheon (Gyeongsangbuk-Do, Republic of Korea) and *A. fulica* was purchased as eggs from a commercial snail farm in Incheon, Korea in 2020 and reared in an insect rearing room at Gyeongsang National University at 60–70% relative humidity (RH) and 24 °C under a 16:8 h [L:D]. For *L. mali*, female adults laid eggs on 2% agar plus water media in insect breeding dish (100 mm dia. × 40 mm). Duration of larvae and pupae were 5–6 days and 25 days, respectively. *Pleurotus eryngii* was supplied as a food source for *L. mali* larvae. The last two instars of *L. mali* larvae (third and fourth instar) were used in this study. *Achatina fulica* juveniles hatched from eggs were maintained on lettuce diet and subjected for fumigation treatments three weeks after hatching.

### 2.2. Fumigants and Imported Orchids

MB was supplied by the Animal and Plant Quarantine Agency (Gimcheon, Republic of Korea). EF (FumateTM, >99% purity) was supplied by Safefume Co., Ltd., (Hoengseong, Republic of Korea). Four species of imported orchids (*C. sinense*, *C. goeringii*, *P. aphrodite* and *A. attenuata*) were purchased from orchid importers in Incheon, Republic of Korea and were used for scale-up MB and EF fumigation trials and for quality evaluation. Orchid seedlings were imported as bareroot seedlings and treated with MB or EF as bareroot seedlings.

### 2.3. Efficacy of EF and MB against A. fulica and L. mali in Small-scale Laboratory Trials

EF and MB efficacies against *A. fulica* and *L. mali* were evaluated using small fumigation chambers (6.8 L desiccator) as described in Kwon et al. (2021) [19]. For *A. fulica*, 20 juvenile snails (3 weeks after hatching) were set in a Petri dish (50 mm × 15 mm, 0.053 μm screen on top) containing a piece of lettuce (20 juveniles/dish). Three dishes containing the juvenile snails were set in each fumigation chamber and fumigated with EF at 5.0–45.0 g/m^3^ for 4 h at 15 °C or with MB at 10.0–70.0 g/m^3^ for 2 h at 15 °C. For *L. mali*, 20 third- and fourth-instar larvae were set in a Petri dish with a slice of mushroom, then triplicate of the dishes were set in the fumigation chamber. *L. mali* were fumigated with EF at 5.0–60.0 g/m^3^ for 4 h at 15 °C or with MB at 10.0–50.0 g/m^3^ for 2 h at 15 °C. Liquid EF was injected to the filter paper (Whatman No. 1) in the desiccator inlet using a gas-tight syringe (100 µL, Hamilton, NV, USA). EF was evaporated in a few seconds in the desiccators due to the injection of small amounts of 0.03–0.36 g to the filter paper even if EF was injected below the boiling point of 54 °C. MB was applied to the inlet area using a gas-tight syringe (100 mL, Hamilton, NV, USA). A mini-fan (6.5 cm i.d. × 3 cm) was set at the bottom of the desiccator for better air circulation. After fumigation was concluded, the fumigants in desiccators were released for 1 h in a fume hood. The pests were then moved to the rearing room (24 °C, 60–70% RH) for 3 d, after which the mortality of treated *A. fulica* and *L. mali* was determined by visual investigation of movement. All treatments, including the untreated control, were repeated three times. The total numbers of *A fulica* and *L. mali* used in the trials were described in Table 1.

The fumigation duration for EF trials was determined as 4 h, considering the current phytosanitary treatment guidelines for imported fruits in Korea: for banana, 35 g/m^3^ at >13 °C for 4 h, for orange, 70 g/m^3^ at >5 °C for 4 h [4]. Equalizing the fumigation time can increase usability or optimize treatment conditions. For MB, the fumigation time was 2 h, following the current guidelines for imported orchids in Korea (48 g/m^3^ at >15 °C for 2 h) [4].

### 2.4. Determination of EF and MB Concentration and Concentration and Time (Ct) Product

The EF and MB concentrations in the fumigation chambers were checked at 0.1, 1.0, 2.0, 4.0 h and 0.1, 1.0, and 2.0 h, respectively, to calculate the Ct product after fumigant injection into the chambers. This was conducted using a Shimadzu GC 17A gas chromatograph (Shimadzu, Kyoto, Japan) installed with a DB5-MS column (30 m × 0.25 mm i.d. × 0.25 µm film thickness; J&W Scientific, Folsom, CA, USA) and a flame ionization detector (FID). Helium was used as a carrier gas at a flow rate of 1.5 mL/min. The oven, injector and detector temperature were maintained at 100, 250 and 280 °C, respectively. The EF and MB concentrations were calculated based on peak areas using external standards. The calibration curve standards were made by spiking a known volume of liquid EF into a 1 l Tedlar^®^ gas sampling bag (SKC Inc., Pittsburgh, PA, USA).

The Ct products were calculated based on the following equation as described in Ren et al. (2011) [32]:$$\mathrm{Ct}=\sum^{\;}\frac{\left(C_{i}+{\;C}_{i+1}\right)\left(t_{i+1}-{\;t}_{i}\right)}{2},$$
where C = concentration of fumigant (mg/L), t = time of exposure (h), i = order of measurement, and Ct = concentration × time product (g h/m^3^).

### 2.5. Evaluation of EF and MB Sorption in Imported Orchids

The sorption of EF and MB in each of the four individual orchid species was determined using a 5% loading ratio (*w*/*v*) in a 0.275 m^3^ fumigation chamber. EF and MB were applied at 35 and 48 g/m^3^ for 4 and 2 h, respectively, at 15 ± 1 °C. Ct products of EF and MB were calculated as described above. The sorption rate was calculated based on concentration reduction over time using [1 − (C_f_/C_0_)], where C_f_ = the final reading of concentration during fumigation and C_0_ = the first reading of concentration during fumigation.

### 2.6. Scale-Up Test of EF and MB Fumigation for Disinfesting Snails and Flies on Imported Orchids

The scale-up fumigation trials to control *A. fulica* and *L. mali* in imported orchids were conducted first in 30 m^3^ then in 150 m^3^ fumigation tents at Incheon port in Korea. The fumigation dose and duration were determined to achieve LCt_99_ of EF and MB on the pests based on the small-scale laboratory trials and sorption evaluations. Each trial was conducted with a 5% loading ratio (*w*/*v*) of all four species of imported orchids at a 1:1:1:1 ratio. Almost 3000 and 15,000 pots of orchids were placed in 30 m^3^ (4.8 × 3.2 × 2.0 m) and 150 m^3^ (9.7 × 6.4 × 2.4 m) tents (Woolim Co., Ltd., Incheon, Republic of Korea), respectively. EF was used at 35 g/m^3^ for 4 h at 15 ± 1 °C in both 30 and 150 m^3^ fumigation tents. MB was first used at 48 g/m^3^ at >15 °C for 2 h in a 30 m^3^ tent. However, due to insufficient mortality from the first test in a 30 m^3^ tent, for the 150 m^3^ MB trial MB was exposed 1 h longer at 48 g/m^3^ at 15 ± 1 °C for 3 h. For each EF and MB trial, we transferred >200 of juvenile *A. fulica* or third- and fourth-instar *L. mali* to each Petri dish (9.5 cm × 8 cm, 0.053 μm screen on top) containing their respective diets and placed one *A. fulica* and one *L. mali* Petri dish in the top, middle, and bottom parts inside the fumigation tent, respectively. Ct products of EF and MB were calculated based on the concentrations of EF and MB determined at 0.1, 1.0, 2.0 and 4.0 h after the completion of fumigant injection. The EF and MB gas in the fumigation tent was sampled from the vicinity of snail and fly Petri dishes, through gas sampling ports into gas sampling bags (1 L, SKC Inc., PA, USA). Assessments of *A. fulica* and *L. mali* mortality and orchid quality were conducted 3 and 7 d after fumigation treatment, respectively, using the same approaches described above. All treatments, including the control, were repeated three times. The total numbers of *A fulica* and *L. mali* used in the trials are described in Table 2.

### 2.7. Effect of EF and MB Fumigation on the Quality of Imported Orchids

The quality evaluation of EF- or MB-treated orchids was conducted using the imported orchids treated in the scale-up fumigation trials described above. Right after the scale-up fumigation trials were concluded, the orchid seedlings were planted in pots for quality evaluation. *C. sinense* and *C. goeringii* were planted in triplicates in pots (0.7 L) containing a mixture of perlite and potting medium. *P. aphrodite* and *A. attenuata* were planted individually in pots (1.5 L) containing a mixture of cocopeat and potting medium. After 7 days post fumigation treatment, the potted orchids were evaluated for leaf chlorophyll content, leaf hue value, overall leaf damage (leaf browning), and new leaf development. The leaf chlorophyll content was measured using a chlorophyll meter (SPAD502 Plus, Minolta, Tokyo, Japan), the leaf hue value was determined using a colorimeter (TES 135A, Electrical & Electronic Corp., Taipei, Taiwan), and the overall leaf damage was estimated using a damage index [0: no leaf damage, 1: <5% of total leaf area affected, 2: 5–25% of leaf area affected, 3: 25–50% of leaf area affected, 4: 50–70% of leaf area affected, and 5: >70% of leaf area affected]. Additionally, the potted orchids were investigated for recovery from the fumigation damage (e.g., formation of new leaves) 30 days after fumigation treatment. All treatments, including the untreated control, were repeated three times.

### 2.8. Data Analysis

The dose–response effects of EF and MB on *A. fulica* and *L. mali* were estimated through Probit analysis [33]. Differences in EF or MB sorption in imported orchids were analyzed using an independent t-test, and the differences among four species of orchids were assessed using (SPSS ver. 23). Differences in the quality measurements of imported orchids between fumigated vs. untreated controls were estimated using an independent *t*-test (SPSS ver. 23).

## 3. Results

### 3.1. Efficacy of EF and MB against A. fulica and L. mali

The toxicities of EF and MB to *A. fulica* and *L. mali* were not linear. Based on LCt_99_ (lethal concentration x time product required for 99% mortality), EF was more effective at controlling *A. fulica* than *L. mali*, while MB effectiveness was not significantly different at controlling *L. mali* vs. *A. fulica* (Table 1). For achieving EF efficacy, the LCt_50_ and LCt_99_ values were 26.6 and 68.1 g h/m^3^ for *A. fulica* and 43.6 and 73.1 g h/m^3^ for *L.mali* at 15 °C after 4 h EF fumigation, respectively. For achieving MB efficacy, the LCt_50_ and LCt_99_ values were 34.6 and 95.9 g h/m^3^ for *A. fulica* and 32.9 and 78.4 g h/m^3^ for *L. mali* at 15 °C after 2 h MB fumigation, respectively.

### 3.2. Evaluation of EF and MB Sorption in Imported Orchids

The sorption rates of EF and MB in the four imported orchid species (*C. sinense*, *C. goeringii*, *P. aphrodite*, and *A. attenuata*) are shown in Figure 1. EF showed greater sorption by orchids than MB. The average EF sorption rate in the four species of orchids was 43%, while it was 12% (*t* = 22.4, *p* < 0.01) for MB. Different orchid species showed different sorption rates of EF and MB. The average EF sorption in *P. aphrodite* was 49%, which was 9% greater than that in the other three orchid species (*F_3,8_* = 194.6, *p* < 0.01). The average MB sorption in *C. sinense* was 17%, while it was approximately 10% (*F_3,8_* = 47.4, *p* < 0.001) in the other three orchid species.

### 3.3. Scale-Up Test of EF and MB Fumigation for Disinfesting Snail and Fly in Imported Orchids

EF fumigation trials conducted with 35 g/m^3^ EF for 4 h at 15 ± 1 °C in 30 and 150 m^3^ tents resulted in mean Ct values of 86.8 and 84.3 g h/m^3^, respectively, leading to the complete control of treated *A. fulica* and *L. mali* (Table 2). MB trials in a 30 m^3^ tent, based on the current guidelines in Korea (48 g/m^3^ for 2 h at >15 °C), resulted in complete control of *L. mali* but only 96.4% morality of *A. fulica*, which was likely due to the lower Ct value (86.7 g h/m^3^) applied compared to its LCt_99_ value (95.9 g h/m^3^) and upper confidence limit (117.4 g h/m^3^) (Table 1 and Table 2). When MB exposure time was increased from 2 h to 3 h in the subsequent 150 m^3^ tent trials, the same MB dose used in 30 m^3^ trials resulted in a Ct value of 116.0 g h/m^3^, achieving complete control of both *A. fulica* and *L. mali* (Table 2). During fumigation in a 30 m^3^ tent, concentrations of EF and MB decreased, similar to the trend observed in the EF and MB sorption tests (Figure 2).

### 3.4. Effect of EF and MB Fumigation on the Quality of Imported Orchids

The quality evaluation results are summarized in Table 3. While there was no apparent damage to *C. goeringii* from the EF treatment, it induced damage tp *C. sinense*, *P. aphrodite* and *A. attenuata* with overall damage indexes of 1.3, 1.7 and 3.7, respectively. One month after EF treatment, *C. sinense* and *P. aphrodite* recovered by forming new leaves. However, *A. attenuata* did not produce new leaves and died. Damage from the MB treatment was greater than for the EF treatment, with average damage indexes of 3.8, 3.3, 3.2 and 4.2 for *C. sinense*, *C. goeringii*, *P. aphrodite* and *A. attenuate*, respectively. All MB-treated orchids did not recover and died before the scheduled evaluation at 30 d post MB treatment. While no significant differences in chlorophyll content and hue value were observed between EF-treated and untreated control orchids, there were significant differences in chlorophyll content and hue value between the MB-treated and control orchids (for MB, all *p*-values < 0.05).

## 4. Discussion

Our results show that EF fumigation is at least as effective as the currently approved MB fumigation guidelines in Korea for the disinfestation of snails and flies in imported orchids. We also found EF to be less phytotoxic than MB. When the effects of EF and MB fumigation on *A. fulica* and *L. mali* were compared, EF treatment at a LCt_99_ dose (35 g/m^3^ EF for 4 h at 15 °C) resulted in complete control of *A. fulica* and *L. mali* in both 30 and 150 m^3^ scale-up trials, conducted with a 5% loading ratio of imported orchids. In contrast, during the initial scale-up tests in the 30 m^3^ tent, the MB treatment at the currently approved dose for imported orchids in Korea (48 g/m^3^ MB for 2 h at >15 °C) resulted in complete control of only *L. mali,* but not *A. fulica*. This is because the Ct product from the trial (86.7 g h/m^3^) was lower than the 95.9 g h/m^3^ MB necessary for 99% control of *A. fulica*. When the same MB dose was extended to 3 h exposure in 150 m^3^ tent trials, a greater Ct product of 116.0 g h/m^3^ was achieved, which led to the complete control of both *A. fulica* and *L. mali*. Regarding the effects of EF and MB fumigation on orchid damage, although both fumigants generally resulted in some level of leaf browning (with the exception of no leaf damage from EF on *C. goeringii*), EF appeared to cause less damage on the treated orchids than MB. Moreover, the orchids treated with EF showed better recovery from the damage than with MB. In fact, the orchids treated with MB in this study died within 30 d after the MB treatment, while only one species (*A. attenuata*) among the four died within 30 d after the EF treatment. In addition, MB treatment resulted in reduced chlorophyll content and increased hue values for treated orchids, while EF treatment did not affect these measurements. Put together, our results suggest that EF is a feasible MB alternative for treating imported orchids, in terms of similar treatment efficacy and lower phytotoxicity.

The phytotoxicity of EF has been previously evaluated for various nursery plants [16], fruits [22,23,28], vegetables [34] and cut flowers and was suggested to be dependent on the treatment dose, plant species and plant cultivar. For example, when nursery plants were treated with EF, EF phytotoxicity was observed ranging from none (e.g., *Ficus benghalensis*, *Pachira macrocarpa*) to severe (e.g., *Anthurium andraeanum*, *Chamaedorea elegans*) [16]. Our results show similar variations in the orchid damage level from the same EF treatment, ranging from *C. goeringii* experiencing no damage, to *C. sinense* and *P. aphrodite* experiencing some but recoverable damage, to *A. attenuata* showing non-recoverable damage. MB damage (40 g/m^3^ for 2 h at 16 °C) on *Cymbidium* spp. has been previously reported and was also supported by our results with the observation of non-recoverable damage to all four species of MB-treated orchids.

MB is more toxic than EF to humans [25]. However, a previous study showed MB was not more toxic to pests than EF: fumigation trials with *P. citri* adults and eggs showed that EF was at least as effective as MB, indicating that the LCt_50_ of EF and MB were 11.93 and 17.48 g h/m^3^, respectively [23]. The pattern of MB and EF efficacy on snails and flies in this study is similar to that recorded in the previous study, although the target pests were different.

Different fumigants often exhibit different sorption characteristics on different commodities. Thus, it is important to understand fumigant-specific sorption properties on different commodities to precisely calculate the necessary dosage for disinfestation of target pests [15,23,32,35]. EF has been known to have a greater sorption rate than other fumigants such as phosphine [16], and our results suggested that rates of EF sorption may be greater than MB. When EF was tested at a 5% loading ratio of imported orchids, it showed greater sorption than MB, with EF and MB sorption rates calculated at 41–49% and 10–11%, respectively. Nevertheless, the LCt_99_ level dose of EF was still achieved even with seemingly greater EF sorption rates of 41–49% in scale-up trials, suggesting that a 5% loading ratio of orchids may be appropriate for the commercial application of EF to imported orchids.

The majority of the exotic snails and flies intercepted from imported orchids and nursery plants in Korea belong to the genera *Achatina*, *Oxychius*, *Lycoriella* and *Megaselia* [8]. The snail and fly species tested in this study, *A. fulica* and *L. mali,* respectively, are already established exotic species in Korea, and were chosen as representative pests for carrying out this study. Although further research is necessary to directly identify and test for the most tolerant exotic snail and fly species, the following important findings emerged from this study: (1) EF and MB may have similar fumigation efficacies for snails and flies, (2) at the same efficacy level, MB causes more severe damage to imported orchids than EF, and (3) the current 2 h MB treatment guideline for imported orchids in Korea is not optimal for reaching the target LCt_99_ level for snails, suggesting a potential need for the revision to the current MB treatment guideline (e.g., 3 h MB exposure).

In conclusion, EF is suggested as a potential alternative to the currently approved MB treatment for controlling exotic snails and flies in imported orchids in Korea. A limitation of our study is that we only tested the juvenile stages of *A. fulica* and the third and fourth instar larvae of *L. mali*. It is thus not clear whether other life stages of *A. fulica* and *L. mali* and, more importantly, other exotic species of snails and flies intercepted from imported orchids can be effectively treated by EF or MB. Thus, additional studies are required, including the determination of EF and MB efficacy for other life stages of *A. fulica* and *L. mali* and the re-evaluation of phytotoxicity in case more EF- or MB-tolerant life stages exist; testing more intercepted exotic pests for efficacy data and phytotoxicity evaluation; and confirmatory trials using imported orchids infested by exotic snails, flies, or their similarly resistant replacements using >30,000 specimens of the most tolerant pest species or life stages.

## Figures and Tables

**Figure 1 insects-14-00066-f001:**
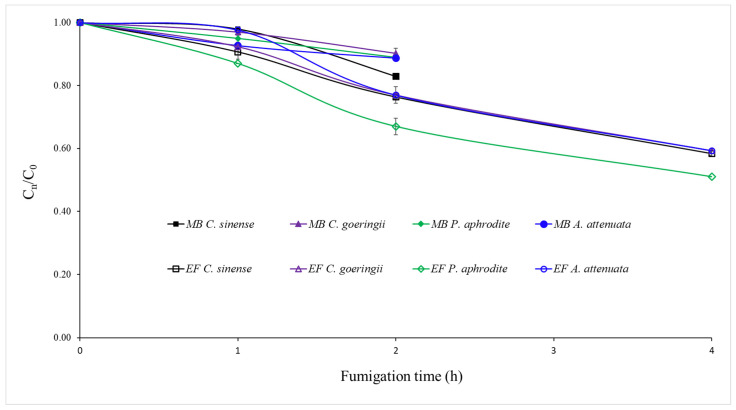
Comparative sorption of ethyl formate (EF) and methyl bromide (MB) in four different species of imported orchids (*C. sinense*, *C. goeringii*, *P. aphrodite* and *A. attenuata*) when EF and MB were applied at 35 g/m^3^ for 4 h at 15 ± 1 °C and 48 g/m^3^ for 2 h at 15 ± 1 °C, respectively, with a 5% loading ratio (*w*/*v*).

**Figure 2 insects-14-00066-f002:**
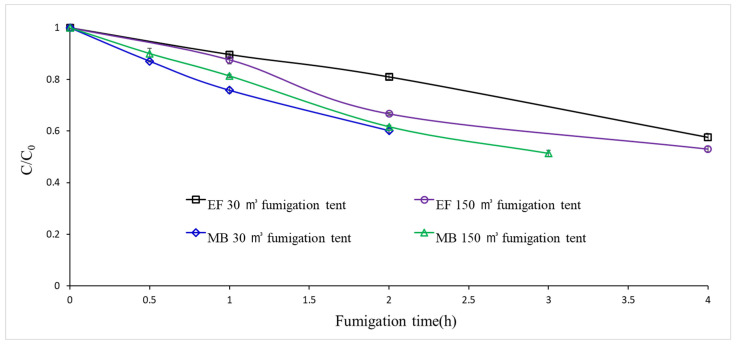
Concentration loss of methyl bromide (MB) and ethyl formate (EF) (C/C_0_) in 30 and 150 m^3^ fumigation tents with a 5% loading ratio (*w*/*v*) of imported orchids. MB was treated at 48 g/m^3^ at 15 ± 1 °C for 2 h during 30 m^3^ tent trials and for 3 h during 150 m^3^ tent trials and EF was treated at 35 g/m^3^ for 4 h at 15 ± 1 °C in both sizes of fumigation tents. EF concentration was checked at 0.1, 1.0, 2.0 and 4.0 h after EF injection to the fumigation tents. MB concentration was checked at 0.1, 1.0 and 2.0 h in 30 m^3^ tent trials and 0.1, 1.0, 2.0 and 3.0 h in 150 m^3^ tent trials after MB injection to the tents.

**Table 1 insects-14-00066-t001:** Lethal concentration ×time (LCt, g h/m^3^) of ethyl formate (EF) and methyl bromide (MB) to juveniles of *Achatina fulica* and third- and fourth-instar larvae of *Lycoriella mali* under 4 h EF fumigation and 2 h MB fumigation at 15 ± 1 °C. CI: confidence interval, SE: standard error.

Fumigant	Pest Species	Number Treated	LCt_50_ (95% CI, g h/m^3^)	LCt_99_ (95% CI, g h/m^3^)	Slope ± SE	*df*	*χ^2^*
EF	*A. fulica*	1800	26.6 (24.9–28.3)	68.1 (60.0–80.2)	5.7 ± 0.4	8	190.1
	*L. mali*	2160	43.6 (37.1–53.3)	73.1 (68.4–100.2)	3.1 ± 0.3	10	28.9
MB	*A. fulica*	1620	34.6 (32.2–37.0)	95.9 (82.7–117.4)	5.3 ± 0.5	7	135.5
	*L. mali*	1440	32.9 (30.3–49.9)	78.4 (72.1–88.9)	2.8 ± 0.7	6	32.6

**Table 2 insects-14-00066-t002:** Concentration × time (Ct) and efficacy of ethyl formate (EF) and methyl bromide (MB) fumigation on *Achatina fulica* juveniles and the third- and fourth-instar larvae of *Lycoriella mali* in scale-up trials (30 and 150 m^3^). In both size trials, EF was treated at 35 g/m^3^ for 4 h at 15 ± 1 °C. For 30 m^3^ MB trials, MB was treated at 48 g/m^3^ for 2 h at 15 ± 1 °C. For 150 m^3^ MB trials, MB was treated at 48 g/m^3^ for 3 h at 15 ± 1 °C. Con and Trt mean control and treated.

Size of Trials (m^3^)	EF Ct Value (g h/m^3^)	MB Ct Value (g h/m^3^)	Insect Pests		Num. Treated with EF	Mortality (%) from EF Treatment	Num. Treated with MB	Mortality (%) from MB Treatment
30 m^3^	86.8 ± 0.7	86.7± 0.7	*A.* *fulica*	Con	602	0.0 ± 0.0	621	0.0 ± 0.0
				Trt	608	100.0 ± 0.0	607	96.4 ± 0.3
			*L. mali*	Con	638	0.0 ± 0.0	652	0.0 ± 0.0
				Trt	611	100.0 ± 0.0	611	100.0 ± 0.0
150 m^3^	84.3 ± 2.2	116.0 ± 0.6	*A.* *fulica*	Con	1137	0.0 ± 0.0	1022	0.0 ± 0.0
				Trt	1115	100.0 ± 0.0	1093	100.0 ± 0.0
			*L. mali*	Con	1269	0.0 ± 0.0	1046	0.0 ± 0.0
				Trt	1207	100.0 ± 0.0	1002	100.0 ± 0.0

**Table 3 insects-14-00066-t003:** Effect of ethyl formate (EF) and methyl bromide (MB) fumigation on the overall damage, chlorophyll contents and hue values of orchids right after scale-up fumigation trials (30 and 150 m^3^). Different letters (a, b) on means indicate significant differences between treated and untreated controls at *p* < 0.05. *n* = 6. – no need to check.

		EF				MB			
Imported Orchids		Damage Index	Chlorophyll Content	Hue Value	Recovery of Treated after 30 d	Damage Index	Chlorophyll Content	Hue Value	Recovery of Treated after 30 d
*C. sinense*	Con	0.0 ± 0.0 ^a^	37.1 ± 1.1 ^a^	13.3 ± 0.8 ^a^		0.0 ± 0.0 ^a^	51.0 ± 2.0 ^a^	16.0 ± 0.5 ^a^	
	Trt	1.3 ± 0.3 ^b^	38.1 ± 0.9 ^a^	14.3 ± 0.4 ^a^	Yes	3.8 ± 0.2 ^b^	35.2 ± 1.3 ^b^	19.8 ± 0.5 ^b^	No
*C. goeringii*	Con	0.0 ± 0.0 ^a^	42.5 ± 0.7 ^a^	19.9 ± 1.2 ^a^		0.0 ± 0.0 ^a^	49.2 ± 1.0 ^a^	17.8 ± 1.1 ^a^	
	Trt	0.0 ± 0.0 ^a^	40.0 ± 1.3 ^a^	20.5 ± 0.7 ^a^	--	3.3 ± 0.2 ^b^	39.5 ± 0.8 ^b^	21.3 ± 0.6 ^b^	No
*P. aphrodite*	Con	0.0 ± 0.0 ^a^	54.5 ± 2.8 ^a^	11.0 ± 0.6 ^a^		0.0 ± 0.0 ^a^	60.4 ± 2.8 ^a^	12.4 ± 1.2 ^a^	
	Trt	1.7 ± 0.2 ^b^	56.9 ± 2.8 ^a^	11.4 ± 0.8 ^a^	Yes	3.2 ± 0.2 ^b^	50.7 ± 0.5 ^b^	20.3 ± 0.5 ^b^	No
*A. attenuata*	Con	0.0 ± 0.0 ^a^	28.7 ± 1.6 ^a^	15.5 ± 0.7 ^a^		0.0 ± 0.0 ^a^	29.6 ± 0.6 ^a^	15.9 ± 0.6 ^a^	
	Trt	3.7 ± 0.3 ^b^	25.5 ± 2.3 ^a^	16.9 ± 0.7 ^a^	No	4.2 ± 0.2 ^b^	22.8 ± 0.7 ^b^	18.6 ± 0.5 ^b^	No

## Data Availability

All the data supporting the findings of this study are available from the corresponding authors upon reasonable request.

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
