# Peer review of "Comparison of Methyl Bromide and Ethyl Formate for Fumigation of Snail and Fly Pests of Imported Orchids"

_insects, 2023, doi:10.3390/insects14010066_

Round 1

Reviewer 1 Report

This manuscript compared fumigation treatment between EF and MB on exotic pest species of snails and flies, and the results are suggesting potentials of EF as alternative to MB which is phasing out globally. The results were well justified and the manuscript was well organized. I’ve had minor concerns on throughout the text, and I suggest authors to revision based on the comments, listed below, before the final decision can be made. I suggest minor revisions to this manuscript.

Minor revision:

Line 27: This sentence is a little awkward. Please consider: “…in imported orchids that is primarily controlled by methyl bromide……”

Line 44: Insert “-” between the “19952015”

Line 72: Delete “to have”

Line 96: Clarify the “(6.8 1)”

Line 107: Italicize “L. mali”

Line 108: Change “from egg hatch” to “after hatching”

Line 135-136: Were new lettuce and muchroom provided for tested snails and flies? There might be risks of those fumigated food poisoning the snails and flies by ingestion if not changed with fresh food after treatment.

Line 138-139: Change “performed in triplicate” to “each repeated three times”, revise this in line 180-181, 198-199 as well. Also, provided the total number of the snails and flies used for the tests.

Line 142: change the text to “…0.1, 1.0, 2.0, 4.0h, and 0.1, 1.0, 2.0h, respectively, …”

Line 162: What is “Cf=xxxxx, C0=xxxxx”?

Line 169: Delete “, when treated”

Line 172-173: How were the petri dishes treated? Were they loosely closed or kept open?

Line 194-196: Is this empirical experience or based on reference? If so for the later, provide the reference.

Line 200: How were the mortality data analyzed prior to probit analysis? Were there any conversion or normality test, such as arcsin square root conversion conducted? Also, were there Abbott correction done prior to the analysis?

Line 265: what did you mean by “regrew”? Does this mean the leaves came off due to the treatment and then recovered by growing back?

Line 287: Change “and not” to “but”. Is the insufficient efficacy possibility due to lack of air circulation inside the chamber?

Line 314: “ondifferent” to “on different”

Line 312-322: I am curious about whether there’s been tests on desorption of EF treated commodities?

Line 339: “byEF” to “by EF”

Reviewer 2 Report

Actuality: The European Green Deal prohibits the use of MBs, so research into the effectiveness of alternative safeguards is globally relevant. 

The title should more accurately reflect the essence of the study, but at this point even "Comparison of methyl bromide and ethyl formate for fumigation of snail and fly pests in imported orchids"  is partially incorrect, because snails are not found inside the plant. L.3. Moreover, three things have been studied: fumigation of snail and fly pests on imported orchids and EF and MB sorption in orchids. There is another problem too - in this study was studied the effectiveness of certain doses and duration and these doses were different.  It turns out that MB was tested at the dose specified by the current fumigation guidelines for imported orchids (48 g/m3, 2 h at >15 91 ℃) and EF was tested at LCt99 (lethal concentration x time product required for 99% 92 mortality) dose determined for A. fulica and L. mali. (L.90-91) Neither the title nor the abstract or simmple summary and  reveal this clearly, why concentrations were different. In other words, it is not clear why the efficiency of different doses (and different exposure time) of EF and MB is compared. It is clear that the general effectiveness of these measures is different (if we take the same doses and duration of exposure), so it is not completely clear why the results obtained at different doses and time are compared with each other. 

Introduction: 

L.44 ,,between 19952015";

L. 61 (48 g/m...

It is strange that MB, which is allowed to be used with an exception, is so toxic to plants and already in the Introductory part it is stated that ,,it causes irreversible damage to orchids (personal observation, THK)". Since this is unpublished data, and this article partially investigates, it should not be discussed in the introduction (L. 59-61).

Materials and Methods:

L.107 L. mali - italic

L.122-123 Kwon et al. (2021) write only about L. mali. (EF and MB efficacies against A. fulica and L. mali were evaluated using small 122 fumigation chambers (6.8 l desiccator) as described in Kwon et al. (2021).)

L.162: ? Cf = xxxxx, C0=xxxxx

Results: 

L. 230-234 Figure 1. Curves overlap, cannot be seen, need to be changed fully.

L. 248-251 Table 2. Why are different doses being compared if EF and MB are known to have different toxicities? It is not explained what it is ,,Con" and ,,Trt" The same - Table 3

L.252-260 Figure 2. Why was the fumigation time stopped after 2 h in one case, after 3 h in another, and continued for 4 h in other cases. In this situation, only fumigation of the same time interval should be compared (i.e. EF 30㎥ fumigation tent and EF 150 ㎥ fumigation tent  and etc.)

L. 266 ,,Damage from the MB treatment was 266 greater than the EF treatment.." I don't understand what else could have been expected and what this conclusion is worth: after all, MB was known to be more toxic, and in this case its dose was higher. Perhaps it is necessary to clarify this here - since it was known that MB is more toxic, higher doses of MB than EF led to a higher effect on plants.

Discussion:

L.306 [15]

L.314  ondifferent

References:

Why are some journals written in full, while others are not? For example: L.356 and L376?  Need to unify all.

L.371, L.374 ,,H.-h." - Does it really have to be lowercase?

L.415  ,,Toxicology & Pharmacology" - Is the hyphen in the text really correct?

Reviewer 3 Report

Dear authors , comments inside of document.

Author Response

Response to Reviewers’ Comments

The manuscript has been rechecked and the necessary changes have been made in accordance with the reviewers’ suggestions. The responses to all comments have been prepared and attached herewith. The manuscript has been revised with the “Track changes” feature on.

Reviewer #3

  1. Line 44, 61, 107: 199252015 to L.mali

Response: Thank you very much for your comments. We have modified them in the revised version.

  1. Line 142: Why not 4 hrs for MB? By pre-test?

Response: Thank you very much for your comments. We have added one paragraph in the last part of 2.3. for clarification.

“The fumigation duration for EF trials was determined as 4 h, considering the current phytosanitary treatment guidelines for imported fruits in Korea: for banana, 35g m-3 at >13 ℃ for 4 h, for orange, 70g m-3 at >5 ℃ for 4 h. Equalizing the fumigation time can increase the usability or optimize treatment conditions in the commercial application by as in previous guidelines. For MB, fumigation time was determined as 2 h, under the current guidelines for imported orchids in Korea (48 g/m3 at >15 ℃ for 2 h)”

  1. Line 209: clarification on LCt, and on sensitiveness on the pests under the CI.

Response: Thank you for your comment. We modified the sentence as follows:

“Based on LCt99 (lethal concentration x time product required for 99% mortality), EF was more effective at controlling A. fulica than L. mali, while MB effectiveness was not significantly different at controlling between L. mali and A. fulica (Table 1)”

  1. Table 1 does not show one value. 4 h EF fumigation and 2 h MB fumigation are different.

Response: Thank you for your comment. The table is shown as one value of LCt, which includes concentration and time. It equalizes concentration and time as one value by calculating how long the pests are exposed at certain concentrations. The reference is as follows:

Reference 23 in this manuscript, Park, M.-G.; Park, C.-G.; Yang, J.-O.; Kim, G.-H.; Ren, Y.; Lee, B.-H.; Cha, D.H. Ethyl Formate as a Methyl Bromide Alternative for Phytosanitary Disinfestation of Imported Banana in Korea With Logistical Considerations. Journal of Economic Entomology 2020, 113, 1711-1717, doi:10.1093/jee/toaa088.

In Table 2, the mortality was compared between ethyl formate (treated at 35 g m3 for 4 h at 13°C) and methyl bromide (treated at 48 g m3 for 2 h at 13°C) although fumigation durations were different.

  1. Legends in Figure 1 should be revised.

Response: Thank you for your comment. The legends in Figure 1 has been modified in the revised version.

  1. Con and trt in Table 1 should be clarified.

Response: Thank you for your comment. We added one sentence in the caption of the table as follows;

“Con and Trt mean control and treated.”

  1. italic in References.

Response: Thank you for your comment. They would be modified in the final step by deleting Endnote style.